# Quantification of cytosolic interactions identifies Ede1 oligomers as key organizers of endocytosis

Dominik Boeke[1,2], Susanne Trautmann[2], Matthias Meurer[2], Malte Wachsmuth[1], Camilla Godlee[1], Michael Knop[2,*] & Marko Kaksonen[1,**]

## Abstract

Clathrin-mediated endocytosis is a highly conserved intracellular trafficking pathway that depends on dynamic protein–protein interactions between up to 60 different proteins. However, little is known about the spatio-temporal regulation of these interactions. Using fluorescence (cross-)correlation spectroscopy in yeast, we tested 41 previously reported interactions *in vivo* and found 16 to exist in the cytoplasm. These detected cytoplasmic interactions included the self-interaction of Ede1, homolog of mammalian Eps15. Ede1 is the crucial scaffold for the organization of the early stages of endocytosis. We show that oligomerization of Ede1 through its central coiled coil domain is necessary for its localization to the endocytic site and we link the oligomerization of Ede1 to its function in locally concentrating endocytic adaptors and organizing the endocytic machinery. Our study sheds light on the importance of the regulation of protein–protein interactions in the cytoplasm for the assembly of the endocytic machinery *in vivo*.

**Keywords** Ede1; endocytosis; fluorescence (cross-)correlation spectroscopy
**Subject Categories** Membrane & Intracellular Transport; Quantitative Biology & Dynamical Systems
**Mol Syst Biol. (2014) 10: 756**

## Introduction

Clathrin-mediated endocytosis is a major, conserved route for the internalization of plasma membrane proteins and extracellular material. Currently, up to 60 proteins have been associated with direct functions in the formation of cargo-containing clathrin-coated vesicles. A hall-mark of endocytosis is the complex orchestration of protein–protein interactions during the various molecular steps in endocytosis. These steps include the recruitment of protein and membrane cargo, the assembly of the endocytic coat, actin polymerization, membrane invagination, scission of the vesicle, and concomitant disassembly of the endocytic machinery, which is then reassembled at new endocytic sites (Kaksonen *et al*, 2006; Ungewickell & Hinrichsen, 2007; Weinberg & Drubin, 2012).

Great advances in understanding the detailed molecular mechanisms underlying endocytosis have come from studies using *Saccharomyces cerevisiae*. It has been shown that the uptake and internalization of cargo, but not the formation of endocytic vesicles, depends on the early endocytic machinery (Brach *et al*, 2014). During the formation of the early endocytic coat, adaptor proteins selectively recognize and bind cargo, lipids, and endocytic coat proteins and link the forming coat to the plasma membrane (Reider and Wendland, 2011; Maldonado-baez *et al*, 2008). To concentrate cargo, these adaptors need to be clustered at the endocytic site. It has been suggested that this clustering depends on a multitude of weak protein–protein interactions between the adaptor proteins, cargo molecules, clathrin, and the early endocytic scaffold protein Ede1 (Maldonado-baez *et al*, 2008). Through its EH domains, Ede1 can bind Asn-Pro-Phe (NPF) motifs (Miliaras & Wendland, 2004). Multiple copies of this motif can be found in various endocytic adaptors, including Yap1801/2 and Ent1/2 (Aguilar *et al*, 2003; Maldonado-baez *et al*, 2008), which in addition have N-terminal lipid-binding domains.

Cargo recruitment is followed by the formation of the late endocytic coat, which is comprised of members of the heterotrimeric Pan1/Sla1/End3 complex and multiple endocytic adaptors. Pan1 has been reported by biochemical assays to physically interact with several other proteins, including Ent1, Yap1801/2, and Sla2, and is therefore a key factor in the organization of the later endocytic coat (Tang *et al*, 1997, 2000; Wendland & Emr, 1998; Wendland *et al*, 1999; Duncan *et al*, 2001; Toshima *et al*, 2007). The Pan1/Sla1/End3 complex has been proposed to cycle between assembled and disassembled states. The interaction between these proteins is negatively regulated via phosphorylation by two redundant kinases Prk1 and Ark1, which are recruited to the endocytic site by the actin-binding protein Abp1 (Cope *et al*, 1999; Zeng *et al*, 2001; Sekiya-Kawasaki *et al*, 2003). In reverse, dephosphorylation via the catalytic phosphatase subunit Glc7 and its adaptor protein Scd5 is required for the Pan1/Sla1/End3 complex to reassemble during a new round of endocytic vesicle formation (Zeng *et al*, 2007).

1 European Molecular Biology Laboratory (EMBL), Heidelberg, Germany
2 Zentrum für Molekulare Biologie der Universität Heidelberg (ZMBH), Deutsches Krebsforschungszentrum (DKFZ), DKFZ-ZMBH-Allianz, Heidelberg, Germany
  *Corresponding author. Tel: +49 6221 544213; Fax: +49 6221 545893; E-mail: m.knop@zmbh.uni-heidelberg.de
  **Corresponding author. Tel: +49 6221 3878285; Fax: +49 6221 3878512; E-mail: kaksonen@embl.de

While the late endocytic coat is forming, proteins of the actin regulation module (WASP/Myo module), including Las17, Myo3/5, and Bbc1, arrive at the endocytic site. Las17 is the main Arp2/3 activator and is regulated by at least four proteins: Sla1, Bzz1, Bbc1, and Syp1 (Soulard *et al*, 2002; Rodal *et al*, 2003; Boettner *et al*, 2009). The arrival of the proteins of the WASP/Myo module is closely followed by the appearance of proteins of the actin filament network, including Actin, Sac6, Abp1, and the proteins of the Arp2/3 complex. They function together to form a branched actin network at the endocytic site, which drives vesicle invagination. During the inward movement of the membrane, the two yeast amphiphysin-like proteins Rvs161 and Rvs167 are simultaneously recruited to the endocytic site and have been proposed to regulate the scission of the vesicle (Kaksonen *et al*, 2005; Kukulski *et al*, 2012).

Fluorescent live cell imaging studies of endocytic events have revealed the regulated and dynamic recruitment of various proteins to the endocytic site and indicated a well-orchestrated hierarchy of arrival and disassembly (Merrifield *et al*, 2002; Kaksonen *et al*, 2003, 2005). The endocytic machinery breaks down at the end of an endocytic event, and the individual components, present in the cytoplasm, are reused to form new endocytic sites (Fig 1A). The regulation of protein–protein interactions outside of these endocytic events is not well understood so that it is currently unclear whether the necessary endocytic components are present as preassembled complexes in the cytoplasm and whether their interactions are regulated at the endocytic site.

In order to study the presence of preassembled endocytic complexes, we used fluorescence correlation spectroscopy (FCS) and fluorescence cross-correlation spectroscopy (FCCS) to investigate protein–protein interactions between the endocytic proteins in the cytoplasm. FCS and FCCS are used to analyze signal fluctuations derived from fluorescently labeled molecules diffusing through a small defined observation volume (Bacia *et al*, 2006; Kim *et al*, 2007). Applied in living cells using fluorescent protein-tagged molecules, it enables the quantification of concentrations, diffusion properties, and oligomerization status of the labeled molecules in the soluble fraction of the cellular protein pools under native and undisturbed conditions. Labeling two different proteins with two spectrally distinct fluorophores in the same sample allows for the assessment of co-diffusion of two proteins and thereby the quantification of protein complexes by FCCS.

Using FCS, we systematically quantified the abundance and diffusion coefficients of 36 endocytic proteins. Using FCCS, we tested for the cytoplasmic presence of 41 protein–protein interactions reported in the literature. Among the 16 cytoplasmic

interactions that we detected, we identified cytoplasmic oligomers of the scaffold protein Ede1, which, like its mammalian homolog Eps15, has a key function in organizing the early stages of endocytosis. We analyzed how the ability of Ede1 to oligomerize contributes to its function in increasing the local concentration of adaptors at the endocytic site. Altogether our approach makes use of quantitative mobility measurements to identify key protein–protein interactions and places them into the context of the dynamic and highly regulated endocytic system.

# Results

## Diffusion coefficient and cytoplasmic concentration of endocytic proteins

To monitor the cytoplasmic concentrations, diffusion coefficients, and protein–protein interactions of endocytic proteins, we used FCS and FCCS. We chromosomally tagged 36 endocytic proteins at their C-terminus with triple tandem fusions of the yeast codon-optimized monomeric eGFP (3myeGFP) or of the yeast codon-optimized mCherry (3mCherry), using mating type *MATa* or *MATα* cells. The triple tandem fusions of the fluorophores were chosen to improve the signal-to-noise ratio in FCS/FCCS measurements (Maeder *et al*, 2007).

To confirm the correct fusion of the triple tags to the proteins, we validated the size of the tagged proteins by Western blotting. In addition, their characteristic localization to cortical patches was confirmed by light microscopy. In order to test the functionality of the fluorescently labeled endocytic proteins, we used assays that score for growth phenotypes of the tagged strains in relation to the corresponding deletion strains and a wild-type strain under standard (30°C, YPD) or stress conditions (37°C or 1 M NaCl). Apl1, Arc18, Las17, Pan1, and Rvs161 could either not be tagged with 3myeGFP or their tagging with 3myeGFP led to a strong growth defect. These proteins were fused to single myeGFP instead. Under stress conditions, phenotypic growth defects were still seen in Rvs161-3mCherry (at 37°C and 1 M NaCl), Rvs167-3mCherry, Rvs161-1myeGFP, and Arc18-3mCherry (all at 1 M NaCl), while these strains behaved normally under the standard conditions used for imaging and FC(C)S.

Using FCS, we measured cytoplasmic diffusion coefficients and cytoplasmic concentrations of 36 proteins and of 1myeGFP and 3myeGFP alone as references for diffusion coefficients (Fig 1B and C, and Supplementary Table S1). The concentration range extended over nearly two orders of magnitude, from

---

**Figure 1. Cytoplasmic concentration and diffusion of endocytic proteins.**

A  Endocytic proteins are disassembled from the endocytic site or the maturing vesicle, released into the cytoplasm, and subsequently recruited to newly forming endocytic sites. The FCS/FCCS observation volume in this study is positioned in the cytoplasm of living yeast cells so that only the cytoplasmic pool of the fluorescently labeled proteins is investigated.

B  Plot of the FCS-determined diffusion coefficients of labeled endocytic proteins, 1myeGFP, and 3myeGFP. Error bars represent the standard deviation derived from single cell measurements (Supplementary Table S1). The average diffusion time is calculated using the half-width of the autocorrelation curve. Proteins within a functional module are sorted from left to right according to their diffusion coefficient values. The color of the bars reflects the functional module that the respective protein belongs to (as indicated in A): green = coat components; red = actin cytoskeleton regulators; blue = amphiphysin module.

C  Plot of the cytoplasmic concentration of endocytic proteins as quantified by FCS. Concentration is calculated using the amplitude of the autocorrelation curve. Error bars represent the standard deviation. Color scheme of the bars is the same as in (B). For details, see also Supplementary Table S1.

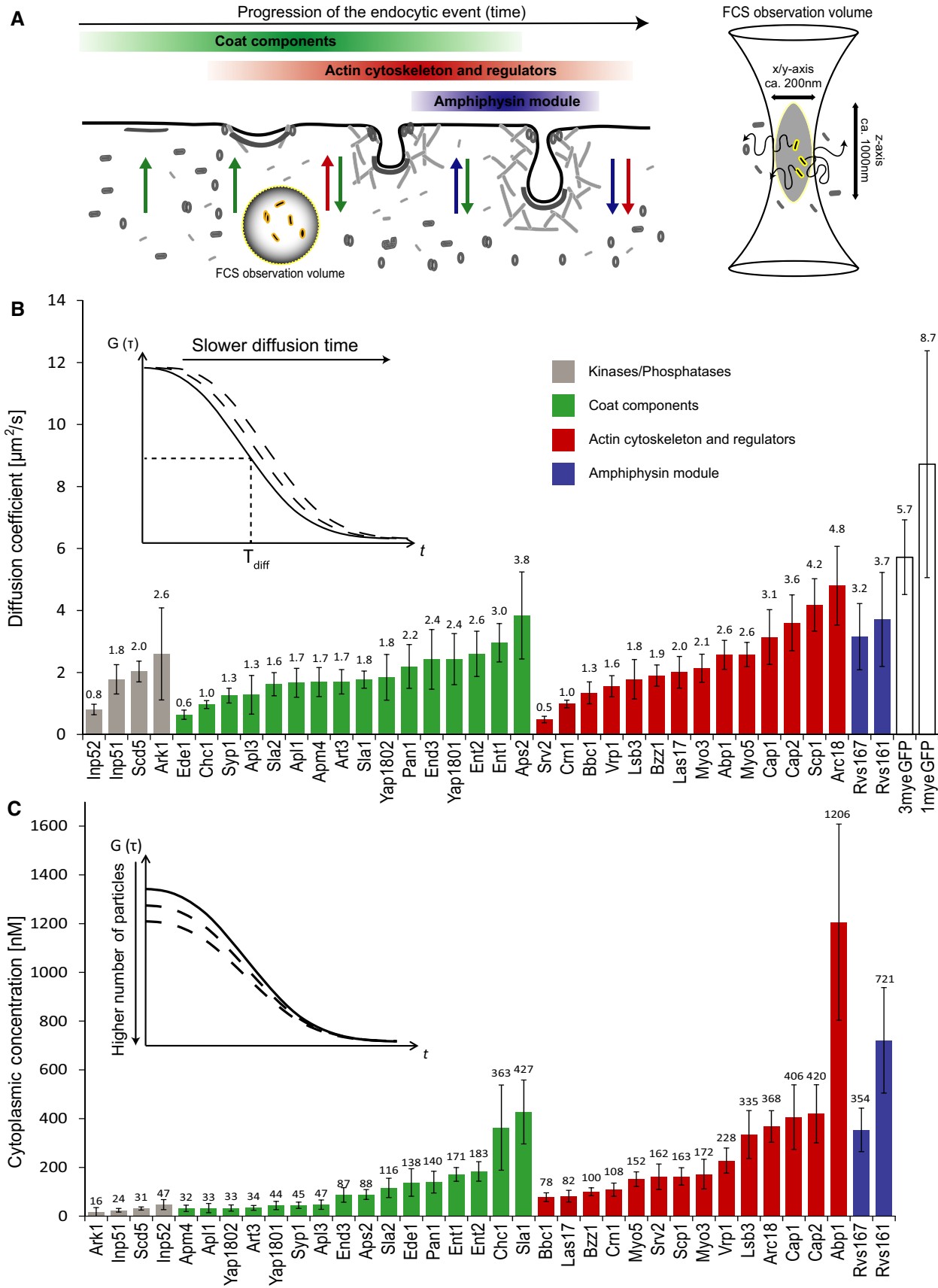

**Figure 1.**

16 nM for the lowest abundant cytoplasmic protein (Ark1) to 1.2 μM for the highest abundant protein (Abp1). Assuming a total cell size of 70 μm$^3$ for haploid *S. cerevisiae* cells (Sherman, 2002) with a cytoplasmic fraction of 29%, as measured for *Schizosaccharomyces pombe* (Wu & Pollard, 2005), the lower and upper concentration values correspond to 195 and 14,625 cytoplasmic molecules per cell. The analysis revealed that regulatory components, including kinases (Ark1, Prk1) and phosphatases (the lipid phosphatases Inp51 and Inp52 as well as the protein phosphatase adaptor protein Scd5) and several endocytic adaptors (Yaps1801/2, Art3, Syp1, and members of the AP2 complex), were present at low levels in the cytoplasm. In contrast, the two yeast amphiphysin-like proteins Rvs161 and Rvs167 and proteins involved in the regulation of the actin cytoskeleton (Abp1, Cap1/2, Arc18) were highly abundant. The diffusion coefficient varied by a factor of ten from 0.48 μm$^2$/s for Srv2 to 4.8 μm$^2$/s for Arc18 (Fig 1B). 1myeGFP and 3myeGFP alone diffused faster with values of 8.7 and 5.4 μm$^2$/s, respectively. The diffusion coefficient for 1myeGFP was comparable to a value of 11 μm$^2$/s measured for single GFP in the cytoplasm by an earlier study (Slaughter *et al*, 2007).

Proteins forming large complexes would be predicted to exhibit slower diffusion compared to monomeric proteins. Among the proteins showing slow diffusion were three members of the AP2-complex. The AP2 complex is a large and stable complex, consisting of four subunits (Apm4, Apl3, Apl1, Aps2) with a total molecular weight of 267 kDa (Yeung *et al*, 1999). The Aps2 subunit exhibited faster diffusion and higher cytoplasmic concentration, suggesting that the cells express Aps2 in excess over the other AP2 subunits. Other slow-diffusing proteins included Chc1 and Srv2 that form stable trimers or hexamers, respectively (Pearse, 1975; Chaudhry *et al*, 2013), and the early endocytic proteins Ede1 and Syp1. In general, the diffusion coefficients correlated weakly with the molecular weights of the tagged proteins (Supplementary Fig S1), consistent with the idea that most tagged proteins are not part of large protein assemblies.

## Analysis of protein complex formation by FCCS reveals multiple cytoplasmic interactions of endocytic proteins *in vivo*

Interactions between endocytic proteins have been studied extensively in biochemical experiments. To understand whether specific protein complexes break apart upon their disassembly from the endocytic site, we wanted to quantify potential cytoplasmic interactions between these proteins. This would show which previously detected protein–protein interactions are not stable in the cytoplasm and which proteins remain in complex after their disassembly from the endocytic site. We first collected data from reported interactions between a selected set of 17 proteins involved in coat formation, adaptor function, actin regulation, and vesicle scission. These comprised 32 protein–protein interactions along with nine homodimers (Fig 2A and Supplementary Table S2).

In order to study protein–protein interactions in the cytoplasm by FCCS, we constructed strains that simultaneously express both 3myeGFP- and 3mCherry-tagged proteins using high-throughput yeast strain crossing (Tong & Boone, 2007). To study homodimer/oligomer formation, we used diploid strains harboring one allele

fused to 3myeGFP and one allele fused to 3mCherry. We then used these strains to quantify the cytoplasmic interaction strength for 33 protein pairs and to investigate homodimer formation of 13 proteins (Fig 2B and Supplementary Table S3). In addition to described interactions, we also investigated a series of potential interactions that have not been described before.

Our FCCS analysis showed that the endocytic machinery is in large part dismantled in the cytoplasm. From the 32 previously described heteromeric interactions, we detected 13 interactions to exist also in the cytoplasm (Fig 2B). Interactions up to a $K_D^{\text{eff}}$ ~1 μM can be reliably detected by FCCS, whereas weaker interactions are outside the dynamic range of this method. Stable interactions were found between two components of the AP2 complex (Apl1 and Apl3), between the two EH domain-containing proteins Pan1 and End3, and between the two yeast amphiphysin homologs, Rvs161 and Rvs167. Weaker interactions included binding of Ede1 to Syp1 and of Sla2 to Pan1, Ede1, and End3. Moreover, several weaker interactions were found around the main Arp2/3 activator, Las17. The analysis of self-interaction detected strong homomeric cytoplasmic interactions between three of the proteins, Sla2, Bbc1, and Ede1. Oligomerization at the order of dimerization has been described for Sla2 before (Wesp *et al*, 1997; Yang *et al*, 1999; Henry *et al*, 2002), and high-throughput studies have shown Bbc1 and Ede1 to self-interact (Krogan *et al*, 2006; Zhang *et al*, 2009). Self-interaction could not be tested for Rvs167 due to problems with the strain. For the other five tested proteins for which reports on homomeric interactions exist (Apl3, Syp1, Pan1, Sla1, Las17), we did not obtain evidence for significant interactions in the cytoplasm. Ede1 has been suggested to contribute as a scaffold protein to the formation of endocytic sites by organizing the early endocytic coat through the interaction with multiple endocytic adaptors. Besides strong homo-oligomerization of Ede1 ($K_D^{\text{eff}}$ = 127 nM), we detected cytoplasmic interaction between Ede1 and Syp1 ($K_D^{\text{eff}}$ = 227 nM) and Sla2 ($K_D^{\text{eff}}$ = 590 nM), but not with the adaptor proteins Yap1802, Ent1, or End3. The values for self-interactions must be considered as the upper limit of the respective $K_D^{\text{eff}}$-value since FCCS only detects interactions between dimers and oligomers that carry both fluorescent protein-labeled species, but cannot identify complexes carrying species with the same label.

In summary, our FCCS results show that the endocytic machinery is in large part dismantled in the cytoplasm. The individual components in the cytoplasm are the cytoplasmic building blocks that are used for the assembly of the endocytic machinery. Our insights into the assembly states of endocytic protein complexes in the cytoplasm can now be used to study the functional role of these protein–protein interactions.

## The early endocytic protein Ede1 forms higher-order oligomers in the cytoplasm

We decided to further investigate the role of cytoplasmic protein–protein interactions using homo-oligomerization of Ede1 as an example. Ede1 is the main organizer of the early stages of endocytosis, and its self-interaction could be a key property of the protein in its function to cluster endocytic adaptors at the endocytic site. Studying the mechanism and function of the self-interaction will therefore contribute to understanding Ede1's role in the endocytic process.

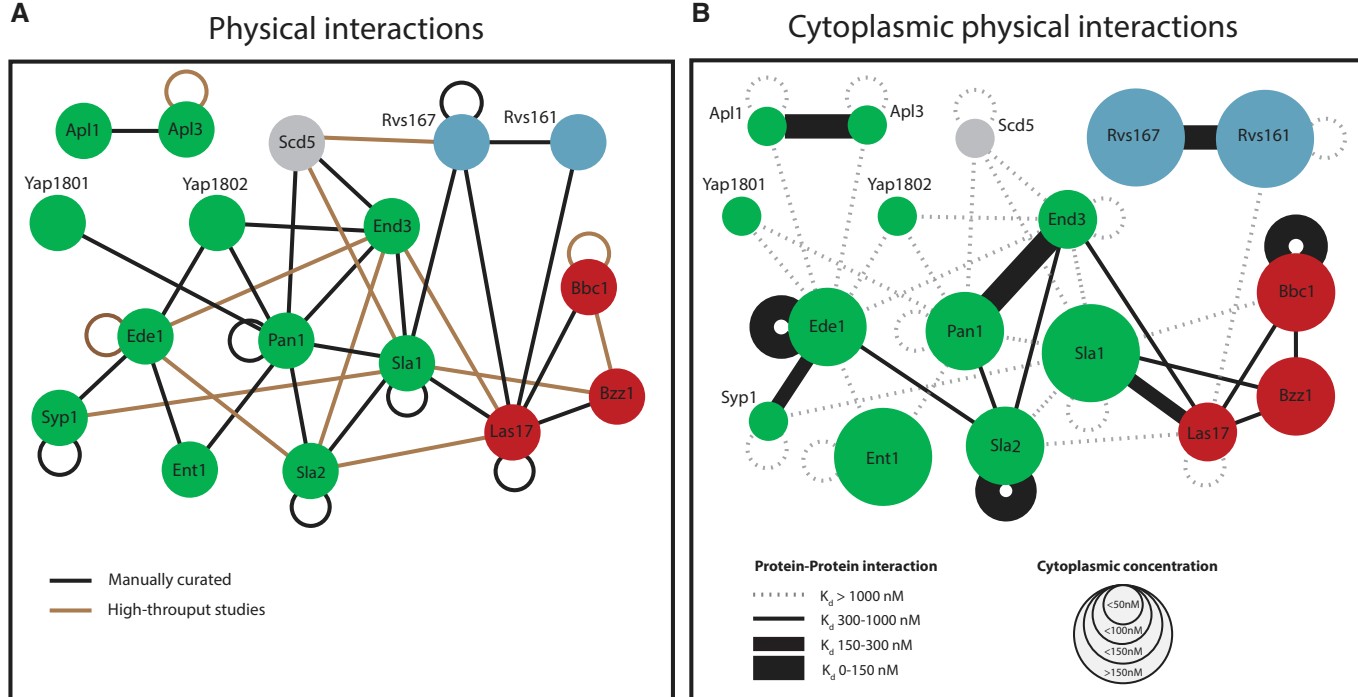

**Figure 2.  Cytoplasmic protein–protein interactions among endocytic proteins measured by FCCS.**

A   Previously reported protein–protein interactions between selected endocytic proteins (see also Supplementary Table S2). Color code of nodes: green = coat components; red = actin cytoskeleton regulators; and blue = amphiphysin module. Loops indicate self-interactions.

B   Network of cytoplasmic interactions based on FCCS data. The size of the nodes represents the cytoplasmic concentration of the protein. Lines between the nodes indicate the protein pairs that were measured by FCCS. A dotted line indicates that no interaction was detected. A solid line indicates an interaction, and the thickness of the solid line represents the strength of the interaction. The expression level of Rvs167-3mCherry was highly variable between individual cells. This was not the case for the GFP-labeled version of the protein, so we only used this fluorescently tagged version. Due to the cross talk from the green into the red channel, we did not test the interaction between Rvs167-1myeGFP and proteins with lower expression levels.

Ede1 showed one of the slowest diffusion coefficients (0.59 μm²/s) of the proteins under investigation, suggesting that it is part of a larger protein complex (Fig 1B, Supplementary Fig S2 and Supplementary Table S1). Self-interaction of Ede1 was also detected by TAP purification of Ede1-GFP in an Ede1-GFP/Ede1-TAP diploid strain (Fig 3A), in consistence with our FCCS data (Fig 2B). To investigate the oligomerization status of Ede1 further, we analyzed the molecular brightness of the diffusing molecular assemblies of Ede1-3myeGFP in counts per particle per second (cpps). In order to calibrate the measurements, we used 3myeGFP and Ent2-3myeGFP to determine the molecular brightness of monomers, and Sla2-3myeGFP as an example for a dimer. As expected for a dimer, Sla2-3myeGFP particles exhibited twice the brightness of 3myeGFP (Fig 3B). The molecular brightness per particle of Ede1-3myeGFP was approximately 2.5-fold higher than that of 3myeGFP and thereby higher than Sla2-3myeGFP. This indicates that Ede1 can form higher-order oligomers. It should be noted that the cpps-value is an average brightness of all detected particles. It does therefore not reveal whether a homogenous population of Ede1 molecules or a dynamic equilibrium between higher order oligomers and monomers is present. The autocorrelation curve of Ede1 could be fitted using a one-component model, suggesting that a possible difference in mass between Ede1 monomers and different Ede1 oligomer populations is not large enough to distinguish them by FCS. In

agreement with Ede1's tendency to oligomerize, we observed the formation of large protein assemblies in the cytoplasm when Ede1 was overexpressed (Fig 3C). Interestingly, when several endocytic adaptors that tether Ede1 to the plasma membrane were deleted, Ede1 weakly localized to endocytic sites and formed similar protein assemblies as in the overexpression strain (Fig 3C). The protein assemblies in this strain are therefore likely to arise from a combination of its tendency to oligomerize and a higher cytoplasmic concentration of Ede1, due to the decrease in recruitment to the endocytic site. The slow diffusion time of Ede1, its high cpps-value, and its tendency to form larger protein assemblies when its cytoplasmic concentration is increased together indicate that Ede1 forms higher order oligomers in the cytoplasm.

### Role of the coiled coil domain of Ede1 in oligomerization

We next asked which part of Ede1 mediates its oligomerization. Besides its unstructured regions, Ede1 contains several protein–protein interacting domains (Fig 4A). The region between amino acids 1,109–1,247 interacts with the μHD domain of Syp1 (Reider *et al*, 2009). This interaction persists in the cytoplasm (Fig 2B). Three N-terminal EH domains interact with NPF motifs found in multiple endocytic adaptors, including Ent1/2 and Yap1801/2 (De Camilli *et al*, 2002; Aguilar *et al*, 2003; Miliaras & Wendland,

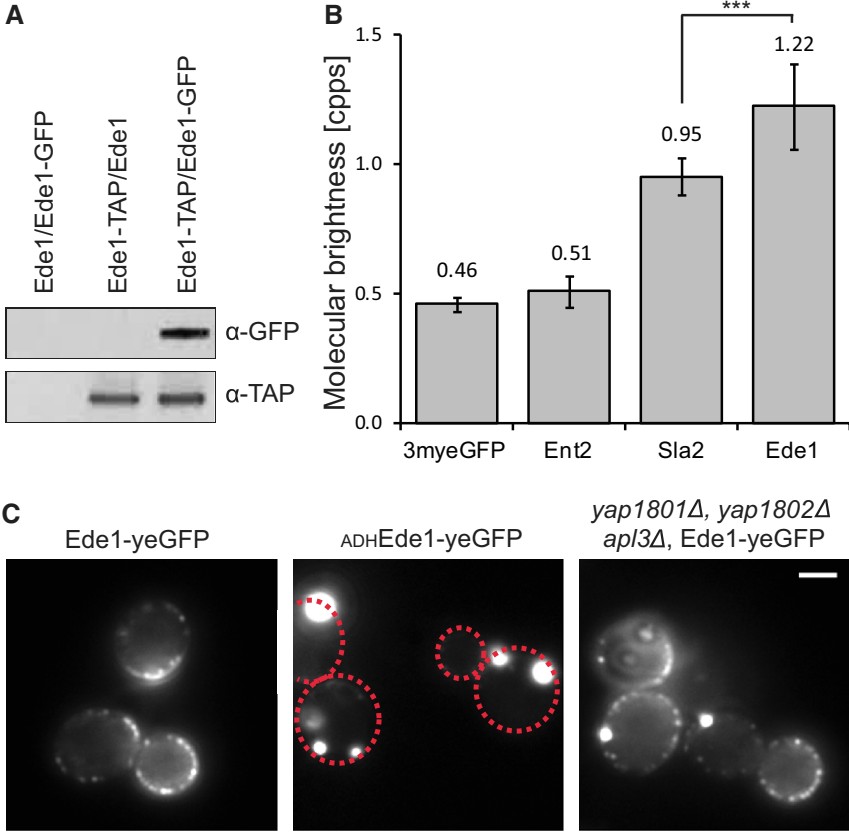

**Figure 3. Cytoplasmic oligomers of Ede1.**

A Tandem affinity purification of a diploid Ede1-TAP/Ede1-GFP strain. Purified proteins were subject to SDS–PAGE and were immunoblotted with α-GFP to detect Ede1-GFP and α-TAP to detect Ede1-TAP.

B Average photon counts per particle per second, measured in 11–14 individual cells per strain. Error bars represent the standard deviation (***P-value ≤ 0.001).

C Fluorescence images of Ede1-yeGFP expressed from its endogenous promoter (left), Ede1-yeGFP expressed from the ADH promoter (middle), and endogenously expressed Ede1-yeGFP in a yap1801, yap1802Δ, and apl3Δ strain (right). Scale bar corresponds to 2 μm. Where needed, dashed red lines were used to outline the cell boundaries.

2004; Maldonado-baez et al, 2008). We did not detect significant cytoplasmic interaction of Ede1 with these proteins by FCCS (Fig 2B). Additionally, Ede1 has a central coiled coil domain and a C-terminal UBA domain for interactions with ubiquitin (Gagny et al, 2000; Shih et al, 2002).

We constructed different endogenously expressed truncation mutants of Ede1 in which we deleted the C-terminal μHD-interacting and UBA domains ($ede1^{\Delta 901-1381}$), the N-terminal EH motifs ($ede1^{\Delta EH}$), or the coiled coil domain alone ($ede1^{\Delta CC}$) or in combination with the C-terminal part ($ede1^{\Delta 591-1381}$) (Fig 4A). Using C-terminal 1myeGFP-fusions, we imaged the different strains by epifluorescence microscopy and calculated the mean fluorescent intensity per pixel in whole cells for $N > 22$. All constructs were expressed at similar levels (Supplementary Fig S3). No obvious difference in localization to wild-type could be observed in the $ede1^{\Delta 901-1381}$ strain (Fig 4A). Deletion of the EH domains did not change localization to endocytic sites, although a pronounced shift in the localization to the bud neck was observed. In contrast, deletion of the coiled coil domain in $ede1^{\Delta 591-1381}$ and $ede1^{\Delta CC}$ resulted in a major loss of the protein's cortical localization. Instead of the distinct localization to the endocytic site, these mutants

localized only weakly to endocytic patches and exhibited a higher cytoplasmic fluorescence. These results suggest a role for the coiled coil domain in endocytic site recruitment.

Strikingly, an ede1-mutant which contained only the coiled coil domain ($ede1^{CC}$) could still localize to cortical patches. This was surprising since all known binding sites for endocytic adaptors were deleted and Ede1 has not been shown to bind membranes directly. However, the residence time of $ede1^{CC}$ at the endocytic site was much shorter than full-length Ede1. We compared the timing of $ede1^{CC}$ recruitment to the endocytic patch with that of Sla1, a marker of the later stages of endocytosis. This revealed that $ede1^{CC}$ shows very similar temporal dynamics to Sla1, with a peak intensity only a few seconds before the peak intensity of Sla1 (Supplementary Fig S4), whereas wild-type Ede1 arrives much earlier than Sla1, as previously described (Stimpson et al, 2009). This suggests that Ede1's coiled coil domain binds to a determinant of the late endocytic stages, while its other domains bind to proteins of the early endocytic machinery, including Syp1 and Yap1801/2. Since $ede1^{CC}$ has very different temporal dynamics, its localization to the endocytic site does not explain the localization of Ede1 to the endocytic site during the early stages of endocytosis. Instead, the coiled coil

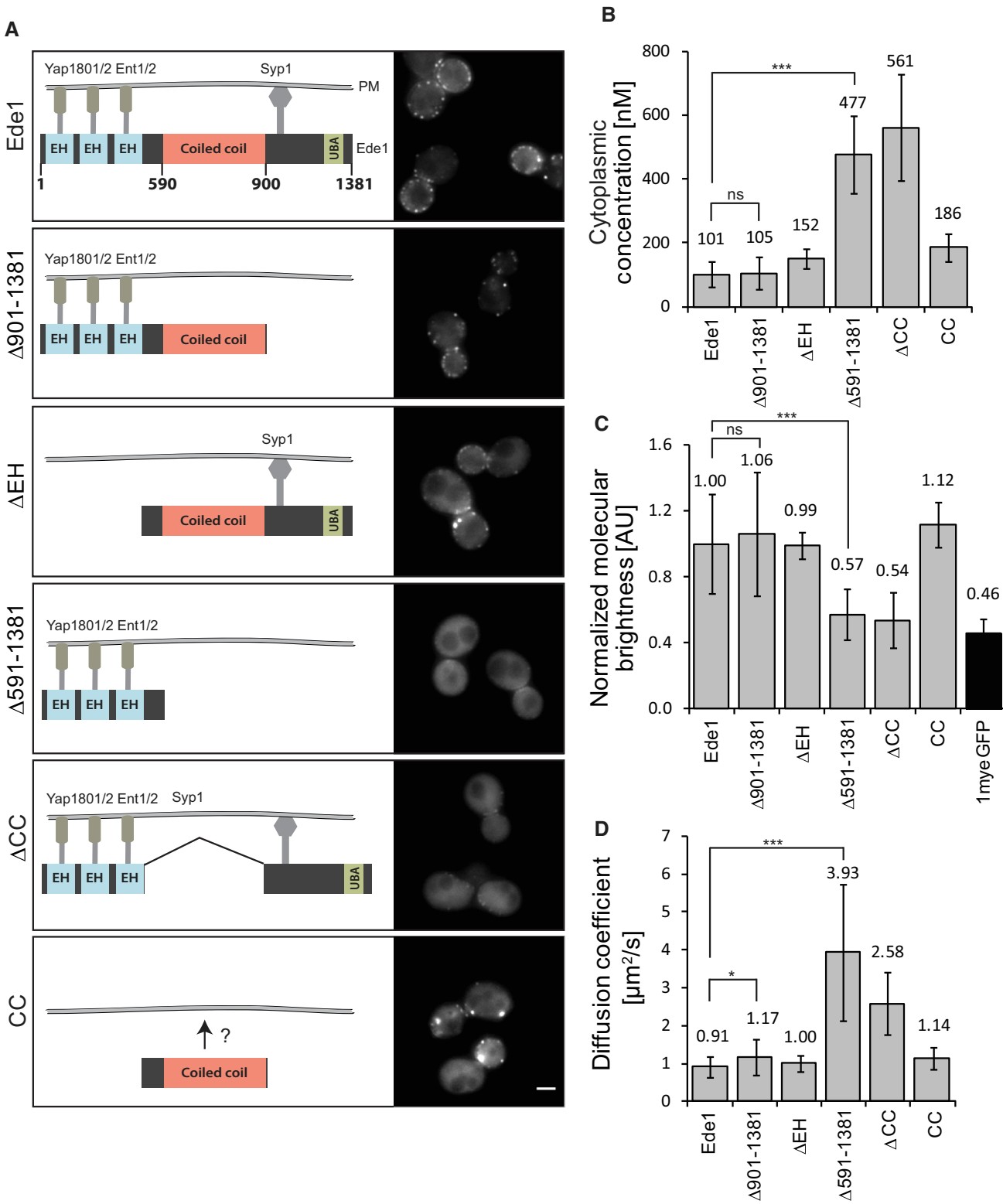

**Figure 4.** ***ede1* mutants lacking the coiled coil domain are mislocalized and cannot oligomerize.**

A    Schematic overview of the domain structure of Ede1 and *ede1* mutants with proposed interaction sites with endocytic adaptors. Domain sizes are schematic and do not exactly match the actual length. The corresponding fluorescence images are shown on the right. Scale bar corresponds to 2 μm.

B–D    Comparison of the cytoplasmic concentration (B), the normalized brightness per particle per second (C), and the cytoplasmic diffusion coefficient of Ede1 and its truncation mutants (D). Measurements were performed in 12–44 individual cells. Error bars represent standard deviation (***$P$-value ≤ 0.001; *$P$-value ≤ 0.05; ns, not significant).

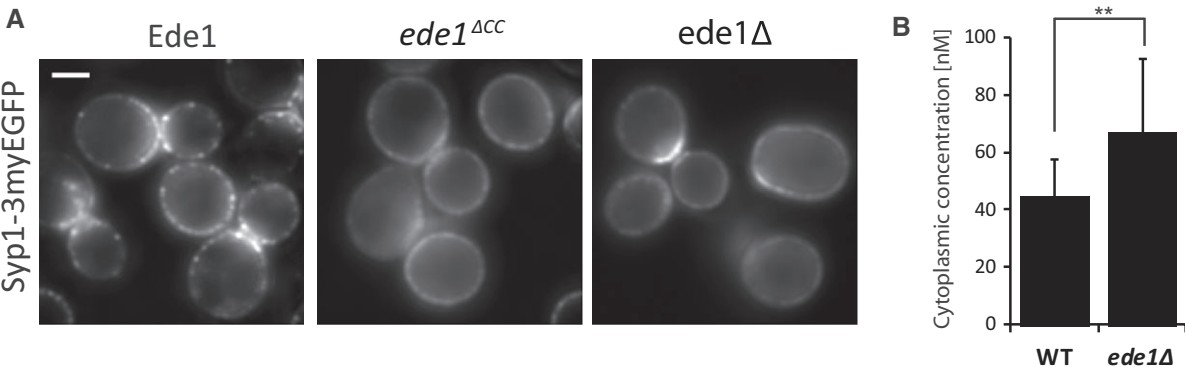

**Figure 5. Mislocalization of the endocytic adaptor Syp1 in ede1 mutant cells.**

A   Average projection of Syp1-3myeGFP in wild-type, $ede1^{\Delta CC}$, and $ede1\Delta$ over a 30-s interval. Scale bar corresponds to 2 µm.

B   FCS measurements of Syp1-3myeGFP in the cytoplasm of wild-type or $ede1\Delta$ cells. Measurements were performed in 12–14 individual cells. Error bars represent standard deviation (**$P$-value $\leq 0.01$).

domain, along with the other protein-binding domains, mediates localization to the early endocytic sites. These results suggest that the coiled coil domain itself possesses another so far unidentified binding site responsible for binding to later endocytic proteins. In addition, $ede1$-mutants that are missing the coiled coil or the EH domains showed abnormal cell shapes, which have been described by an earlier study in $ede1\Delta$ cells (Gagny et al, 2000). We then used FCS to investigate oligomerization of the mutant Ede1 proteins. The cytoplasmic concentration (Fig 4B) and molecular brightness (Fig 4C) of $ede1^{\Delta901-1381}$ were very similar to the wild-type. Also, $ede1^{\Delta EH}$ and $ede1^{CC}$ showed values close to wild-type Ede1, with a slightly increased cytoplasmic concentration and slightly slower diffusion. In contrast, constructs missing the coiled coil domain ($ede1^{\Delta591-1381}$ and $ede1^{\Delta CC}$) showed a strong reduction in molecular brightness, faster diffusion (Fig 4D), and higher cytoplasmic concentration. This suggests that these mutants lost the ability to oligomerize. To confirm this, we constructed a diploid strain expressing $ede1^{\Delta591-1381}$-myeGFP and $ede1^{\Delta591-1381}$-mCherry. This yielded a $K_D^{\text{eff}}$ of > 1 µM by FCCS (Supplementary Fig S5). In addition, no larger protein assemblies in either $ede1^{\Delta CC}$ or $ede1^{\Delta591-1381}$ were detected although these strains showed a higher cytoplasmic concentration of the mutated proteins compared to wild-type Ede1. Furthermore, no co-purification of Ede1$^{\Delta591-1381}$-TAP with full-length Ede1 was observed (Supplementary Fig S5). These data are consistent with the coiled coil domain being responsible for Ede1 self-interaction.

### Ede1-oligomerization underlies early endocytic site organization

The strong mislocalization of $ede1^{\Delta591-1391}$ or $ede1^{\Delta CC}$ is likely to have an effect on their function. Previous studies described that in $ede1\Delta$ cells, endocytic adaptors are not localized in distinct endocytic patches, but are more homogenously distributed over the plasma membrane (Stimpson et al, 2009). We therefore examined whether deletion of the coiled coil domain of Ede1 would lead to a similar phenotype. Syp1-3myeGFP still localized to the membrane in both $ede1^{\Delta CC}$ and $ede1\Delta$ cells. However, it was homogenously distributed over the plasma membrane in these mutants (Fig 5A). The cytoplasmic concentration of Syp1 was increased in $ede1\Delta$ cells as determined

by FCS (Fig 5B). These data show that the coiled coil domain-mediated localization of Ede1 to the plasma membrane is critical for the localization of the endocytic adaptor Syp1.

### Artificial dimerization rescues the localization of $ede1$ mutants

Ede1 mutants missing the coiled coil domain are not efficiently targeted to endocytic sites. The $ede1^{CC}$ mutant on the other hand still localizes to endocytic patches but only during the late phase. The localization of Ede1 to the early stages of endocytosis can therefore not be explained by a binding site in the coiled coil domain. We investigated whether the localization of $ede1^{\Delta591-1391}$ and $ede1^{\Delta CC}$ could be rescued by artificial oligomerization of these mutants. To address this question, we used FRB and FKBP domains, which form a strong heterodimer upon addition of rapamycin (Banaszynski et al, 2005). We constructed a diploid strain, in which one allele of $ede1^{\Delta591-1391}$ was fused to FRB-myeGFP and the other allele of $ede1^{\Delta591-1391}$ to FKBP-myeGFP (Fig 6A). We investigated the effect of rapamycin-induced dimerization on the localization of $ede1^{\Delta591-1391}$ by time-lapse microscopy. Remarkably, within 10 min of rapamycin induction, the mutated protein localized to cortical patches (Fig 6B). This was not seen when only DMSO was added (Supplementary Fig S6). The same effect was observed for artificially dimerized $ede1^{\Delta CC}$. Rapamycin induction led to an increase in photon counts per particle by 32% and a decrease in diffusion coefficient from 3.60 to 1.75 µm²/s (Fig 6C). The cytoplasmic concentration was lower than in untreated mutant cells but higher than in wild-type cells. This suggests that localization was not fully rescued, possibly due to a lack of higher-order oligomers. Almost all of the artificially formed $ede1^{\Delta591-1391}$ patches were followed by the appearance of the endocytic marker Sla1, demonstrating that they are sites of endocytosis (Fig 6D and E). The temporal dynamics of the artificially formed $ede1^{\Delta591-1391}$ patches were similar to the dynamics of wild-type Ede1-1myeGFP patches, showing that these patches mark the early stages of endocytosis (Fig 6E). Importantly, strains harboring $ede1^{\Delta591-1381}$-FRB-myeGFP or $ede1^{\Delta591-1381}$-FKBP-myeGFP alone did not show rescue of the localization of the mutant after rapamycin treatment (Supplementary Fig S6). In summary, these results show that oligomerization of Ede1, either through its

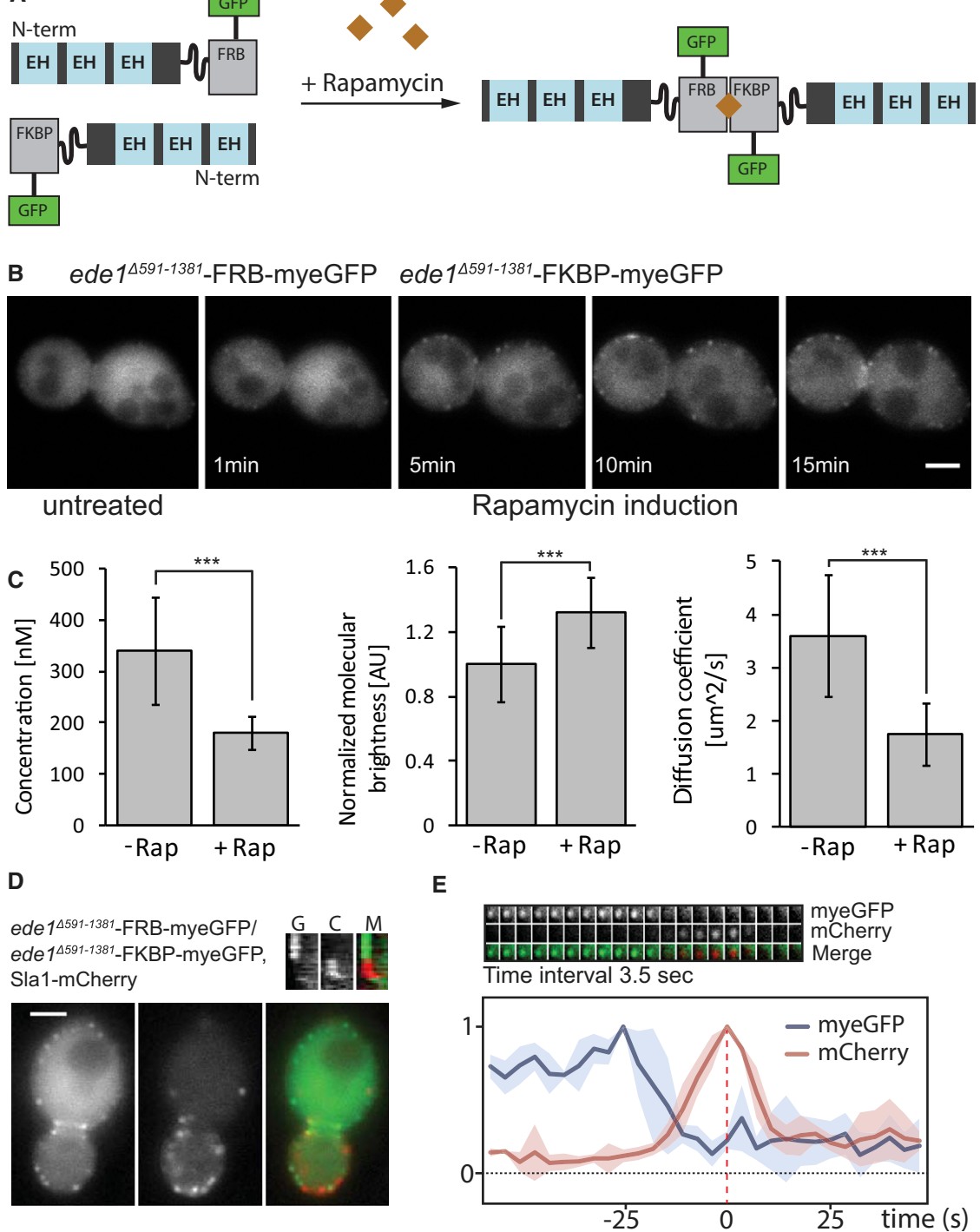

**Figure 6.  Artificial dimerization of ede1 mutants rescues its localization.**

A    Schematic representation of the artificial dimerization of *ede1*$^{\Delta591-1381}$ mutants by rapamycin.

B    Time-lapse microscopy of an *ede1*$^{\Delta591-1381}$-FRB-myeGFP/*ede1*$^{\Delta591-1381}$-FKBP-myeGFP strain before and after rapamycin treatment. Scale bar corresponds to 2 μm.

C    Quantification of the cytoplasmic concentration, normalized brightness per particle per second, and diffusion coefficient for an *ede1*$^{\Delta591-1381}$-FRB-myeGFP/ *ede1*$^{\Delta591-1381}$-FKBP-myeGFP strain before and 30 min after rapamycin treatment (***$P$-value $\leq$ 0.001).

D    Green (left), red (middle), and merge (right) of *ede1*$^{\Delta591-1381}$-FRB-myeGFP/*ede1*$^{\Delta591-1381}$-FKBP-myeGFP, Sla1-1mCherry strain 30 min after rapamycin treatment.  Scale bar corresponds to 2 μm.

E    Top: Time series of a single cortical patch in the same strain 30 min after rapamycin treatment. Time interval between frames is 3.5 s. Bottom: Quantification of the myeGFP and mCherry fluorescence in this strain for four patches plotted as a function of time. Individual intensity curves for myeGFP (blue) and mCherry (red) were normalized independently. For each patch, the myeGFP and mCherry curves were aligned to the peak intensity of Sla1 (= time point 0) in time.

intrinsic coiled coil domain or artificially via FRB and FKBP domains, is needed for the correct localization of the protein to the endocytic site and for its function.

## Discussion

### Monitoring the assembly status of endocytic building blocks in the cytoplasm

In studies elucidating the molecular mechanisms underlying endocytosis, the emphasis has been on understanding the localization of proteins to the endocytic site and their temporal dynamics there. Multiple physical interactions have been shown between endocytic proteins, which together form a large interaction network at the endocytic site; however, most of the available data do not distinguish whether the interactions are constitutive or restricted only to the endocytic site and thus potentially subject to specific regulation. In this work, we aimed at specifically closing this information gap by using FCCS to quantify interactions of endocytic proteins in the cytoplasm. FCCS is ideally suited to address this question, since FCCS measurements are conducted with single-molecule detection sensitivity, which allows the investigation of proteins with low abundances or low cytoplasmic concentration (down to approximately 10 nM, which corresponds to less than 200 molecules per yeast cell). Furthermore, FCCS reports on co-mobility behavior of proteins, which is insensitive to the spatial arrangement of the fluorophores (in contrast to FRET-based methods), and therefore, FCCS can reliably detect the absence of protein–protein interactions within its dynamic range (interactions with $K_D^{eff}$ values < 1 μM). This dynamic range is well suited to identify strong to medium–strong interactions, which are needed to recruit proteins to cellular sites, whereas it is often not sufficient to detect highly transient regulatory interactions ($\gg$ 1 μM). Consequently, the subcellular protein interaction network we generated reports on strong to medium–strong protein–protein interactions in the cytoplasm. Considering that most methods previously used to detect protein–protein interactions, such as protein complex purification or co-immunoprecipitation, also exhibit limited sensitivity toward the detection of weak regulatory interactions (such as kinase substrate or some SH3–peptide interactions), we conclude from our cumulative results that many reported protein–protein interactions are likely to occur only at the endocytic site. Only a few interactions seem to be stable in the cytoplasm. These are discussed in the following sections.

### The cytoplasmic protein–protein interaction network

Biochemical experiments demonstrated interactions between the coat components Pan1, Sla1, and End3 (Tang *et al*, 2000). Some of these interactions are negatively regulated by the action of two redundant kinases, Ark1 and Prk1, which trigger the disassembly of the endocytic machinery by phosphorylation (Zeng *et al*, 2001). Studies investigating the protein interactions among Pan1/Sla1/End3 in the phosphorylated (disassembled) state remained inconclusive. One study proposed that phosphorylated Pan1 is unable to associate with either End3 or Sla1 (Zeng *et al*, 2001). While this would indicate that the Pan1 complex is fully disassembled in a phosphorylation-dependent manner, Toshima and co-workers

suggested that Pan1 and End3 remain in a complex after Prk1-dependent phosphorylation (Toshima *et al*, 2007). Our results demonstrating strong cytoplasmic interaction between Pan1 and End3, but not between Pan1 and Sla1 (or End3 and Sla1), are thus consistent with a model in which only the Pan1–Sla1 (and End3–Sla1) interaction is subject to phosphoregulation. A recent study showing that End3 binds to a region in Pan1 that is not phosphorylated provides further support for the conclusion that the interaction between these two proteins is unlikely to be regulated by phosphorylation (Whitworth *et al*, 2014). Neither Pan1 nor End3 showed a tendency to self-interact in the cytoplasm, suggesting a stoichiometry of 1:1 for the cytoplasmic complex. However, self-interaction/dimerization of Pan1 has been shown by two-hybrid and co-immunoprecipitation experiments (Miliaras *et al*, 2004). Therefore, the Pan1 self-interaction is also likely to be subject to phosphoregulation by Ark1/Prk1. Interestingly, an analogous situation was observed for the MAP kinase signaling scaffold protein Ste5, which only forms dimers or oligomers in its membrane-bound fraction and thereby constitutes an essential part of MAP kinase signaling in the yeast pheromone pathway (Inouye *et al*, 1997; Maeder *et al*, 2007; Zalatan *et al*, 2012). It might be that the regulated self-interaction of Pan1 is a key driver of later stages of endocytosis.

We furthermore observed that the previously reported interaction of Pan1 with Sla2, an inhibitor of the Arp2/3 activation activity of Pan1 (Toshima *et al*, 2007), exists in the cytoplasm (Fig 2). It would be interesting to see whether this interaction also occurs at the endocytic site, or whether it is subject to regulation, for example to restrict Arp2/3-mediated actin nucleation to the site of endocytosis. After disassembly, the recycling of cytoplasmic Pan1 to new endocytic sites is mediated by Glc7 phosphatase via the targeting subunit Scd5 (Zeng *et al*, 2007). Currently, it is unclear when and where this happens: in the cytoplasm, or associated with Pan1 recruitment to new endocytic sites, or both. FCCS did not detect an interaction between Pan1 and Scd5, which indicates that if such an interaction occurs in the cytoplasm, it might be very transient.

We observed that the Wiskott–Aldrich syndrome protein (WASP) Las17, the main Arp2/3 activator, is partly bound to at least three of its regulators, Sla1, Bbc1, and Bzz1, with the strongest interaction between Las17 and Sla1 ($K_D^{eff}$ = 286 nM). At the endocytic site, Sla1 arrives simultaneously with Las17 (Kaksonen *et al*, 2003; Feliciano & Di Pietro, 2012). The inhibitory activity of Sla1 toward Las17 is thought to be relieved upon the subsequent arrival of Bzz1 (Sun *et al*, 2006). Interestingly, we noticed weak but significant cytoplasmic interactions between Bzz1 and Sla1, as well as between Bzz1 and Las17. It remains to be investigated whether these interactions are dependent on the strong interaction of Sla1 with Las17 and how they relate to Bzz1 recruitment to the endocytic sites. Understanding the interplay of these interactions might lead to novel insights into the regulation of actin dynamics. It is interesting to note that our *in vivo*-measured interaction of Sla1 with Las17 has a $K_D^{eff}$ value of 286 nM, whereas the *in vitro*-measured interaction between Sla1's SH3 domains and the polyproline motif of Las17 is significantly stronger ($K_D$ = 56 ± 8 nM; Feliciano & Di Pietro, 2012). This discrepancy might reflect competition for Las17 binding between the three binding partners (Bzz1, Sla1, and also Bbc1; Fig 2), all of which have been suggested to bind Las17 via their SH3 domains. This again indicates an interesting line of investigation toward understanding of the role of these proteins in endocytosis,

where, for example, the effect of selective abortion of individual interactions by other interactions could be monitored by FCCS.

We also investigated the interactions between the two yeast amphiphysins Rvs161 and Rvs167. Mammalian amphiphysins have been shown to form a polymeric structure around the endocytic invagination (Takei *et al*, 1999). In yeast, heterodimerization of these proteins has been shown *in vitro* and *in vivo* and data from a bimolecular fluorescence complementation assay indicated that Rvs167 oligomerizes at the endocytic site (Navarro *et al*, 1997; Friesen *et al*, 2006; Youn & Friesen, 2010). We show that both amphiphysins form a high-affinity heterodimer in the cytoplasm. Moreover, our data are consistent with no homodimerization of Rvs161 in the cytosol, while homodimerization of Rvs167 could not be tested, due to non-functionality of the Rvs167-3mCherry tag. Taken into account the proposed oligomerization at the endocytic site, our data suggest a model in which both proteins get recruited as a heterodimer to the endocytic site upon which the formation of a larger polymer is triggered.

## Oligomerization of Ede1 is essential for its localization and function as an early endocytic scaffold protein

The FCS and FCCS analysis of the scaffold protein Ede1, which is critical for the organization of early endocytic sites, revealed, in contrast to Pan1, strong self-interaction in the cytoplasm. Although our data do not allow a clear distinction between dimers and higher order oligomers, the molecular brightness analysis of the cytoplasmic Ede1-3myeGFP complexes indicated that Ede1 likely exists in oligomeric complexes of more than two Ede1 proteins *in vivo*. In line with this observation, we detected formation of large Ede1 protein assemblies upon Ede1 overexpression. By FCCS, we did not detect significant interaction of Ede1 with Asn-Pro-Phe (NPF) motif-containing adaptor proteins in the cytoplasm (Miliaras & Wendland, 2004) (Fig 2), which is likely explained by the weak and transient nature of the NPF–EH domain interactions with $K_D$-values usually larger than 100 μM (Cesareni *et al*, 2005). Interestingly, in strains containing genomic deletions of multiple adaptor proteins, Ede1 was still able to localize to the endocytic site, but aggregates of Ede1 similar to the overexpression strain were observed. This shows that the deleted adaptors are involved in tethering Ede1 to the membrane and suggests a delicate equilibrium between other Ede1 interactions and the formation of Ede1 oligomers. A further dissection of Ede1 domains indicated that Syp1 and other NPF-containing adaptors seem to have, at least in part, redundant functions in tethering Ede1 oligomers to the membrane and that the coiled coil domain or an adjacent sequence of Ede1 is able to promote Ede1 binding to late endocytic sites, with temporal dynamics similar to Sla1. Binding of Ede1 to early and late endocytic sites may therefore occur through different mechanisms.

## The role of Ede1 in organizing the early endocytic coat

The exact function of Ede1 in organizing the early endocytic coat is not known. The assembly of cargo at the endocytic site has been shown to take place during the time between localization of Ede1 to the membrane and the arrival of the coat component Sla1 (Toshima *et al*, 2006). Deletion of Ede1 results in a more homogenous membrane distribution of endocytic adaptors, including Syp1 (Reider *et al*, 2009; Stimpson *et al*, 2009). In our study, a similar mislocalization of Syp1 was seen in the $ede1^{\Delta CC}$ mutant cells. Since the binding site for Syp1 still exists in the $ede1^{\Delta CC}$ mutant, this indicates that the coiled coil domain contributes to Ede1's *in vivo* function in locally concentrating and stabilizing endocytic adaptors at the endocytic site. Coiled coil domain-mediated oligomerization of Ede1 is likely to increase the avidity between Ede1 and other endocytic adaptors and contribute to the local clustering of them, as illustrated in Supplementary Fig S7. Stabilization and concentration of endocytic adaptors might be an important step in the early endocytic process. Since Ede1, together with Syp1, is the earliest protein to arrive to the endocytic site, we speculate that the cytoplasmic oligomerization of the protein allows a mechanism in which the arrival of Ede1 oligomers triggers the initial clustering of endocytic adaptors. In an $ede1\Delta$ mutant or $ede1^{\Delta EH}$ mutant, the lifetime of early endocytic proteins is significantly altered (Stimpson *et al*, 2009; Suzuki *et al*, 2012), indicating that the early phase of endocytosis functions less efficiently in these strains. It has been suggested that Pan1 and Ede1 have functionally redundant roles in organizing the endocytic coat (Miliaras & Wendland, 2004; Maldonado-baez *et al*, 2008). The oligomerization of Ede1 and stable binding of Pan1 to End3 might indicate that interaction between EH domain-containing proteins is a common mechanism in helping to organize the endocytic coat.

Former studies in mammalian cells indicate that the function of Ede1 as a scaffold protein to locally cluster endocytic adaptors is conserved. Ede1 has four mammalian homologs, Eps15, Eps15R, and intersectin1/2. Interestingly, biochemical assays suggested that Eps15 can form homodimers as well as tetramers through anti-parallel association between two Eps15 dimers (Tebar *et al*, 1996; Cupers *et al*, 1997; Salcini *et al*, 1999). A knockdown of all four homologs affected the clustering of the Syp1 homolog FCHo2 into distinct puncta at the plasma membrane, while their membrane localization per se was not affected, whereas proteins of the AP-2 complex were mostly cytosolic (Henne *et al*, 2010). Through adaptor–cargo interaction, clustering of adaptors is likely to result in clustering of endocytic cargo. The clustering of cargo has been shown to increase the maturation efficiency of clathrin-coated pits in mammalian cells (Liu *et al*, 2010). The knockdown of Eps15 increased the lifetime of clathrin patches, indicating that Eps15 has an important role in the maturation of the endocytic site (Mettlen *et al*, 2009). Besides FCHo2, Syp1 has another mammalian homolog, FCHo1. Both proteins have been proposed to demarcate cell membrane patches for clathrin assembly and to directly recruit Eps15 and intersectin to endocytic sites, which then in turn recruit the AP-2 complex (Henne *et al*, 2010). Our data support a conserved mechanism, in which Syp1 (FCHo1/2) and other endocytic adaptors recruit Ede1 (Eps15/intersectin) oligomers to the endocytic site, which in turn leads to their clustering, in preparation for the formation of an endocytic vesicle.

Studying the structure–function relationship of Ede1 allowed us to directly link the insights obtained from our systematic FCCS screening approach to the mechanistic function of the protein during the process of endocytosis. The subcellular protein interaction network that we generated will therefore not only allow distinguishing between which interactions are subject to spatial regulation but will

also open up the study of the importance of non-regulated versus regulated interactions for an intact endocytic machinery.

# Materials and Methods

### Yeast strains and growth conditions

All yeast strains used are listed in Supplementary Table S4. Standard yeast methods and growth media were used. Strains were grown in standard rich medium (yeast extract/peptone/dextrose; YPD) or synthetic complete medium without tryptophan (SC-Trp) for microscopy. For the experiment shown in Fig 6, dimerization was induced by the addition of rapamycin (Sigma-Aldrich, R8781) at a concentration of 4.5 μg/ml (5 μM) for the indicated times (from a stock solution in DMSO). Since cells are usually sensitive to rapamycin, the *tor1-1* mutation was introduced into the genome. This mutation leads to rapamycin resistance. In addition, the *FPR1* gene was deleted. Endogenous Fpr1 binds rapamycin and would therefore compete for binding to rapamycin with the introduced FRB/FKBP domains. For growth assays, yeast colonies were grown on 96-well plates on YPD and replica-plated and grown under the respective conditions/media.

### Chromosomal manipulations of yeast strains

C-terminal tagging, gene deletions, and promoter substitutions were generated by homologous recombination into the endogenous gene locus as previously described (Janke *et al*, 2004). In all cases, monomeric yeast-enhanced GFP (myeGFP) or mCherry was used for fluorescent tags. Cassettes used for PCR targeting of triple tandem fusion of GFP and mCherry were described earlier (Maeder *et al*, 2007) and were further optimized using different codon usage to avoid homologs' recombination within the cassette. Plasmids are listed in Supplementary Table S5 (further details and sequences available upon request). Correct integration of the cassettes into the genome of yeast strains was tested by PCR and fluorescence microscopy. The expression of full-length tagged proteins with the expected molecular weight was validated using Western blotting.

### Data mining of reported interactions

Reported interactions between the chosen set of proteins were obtained by STRING (http://string-db.org), applying the highest confidence score and taking into account only physical interactions from both high-throughput and manually curated studies. Data about self-interactions were obtained from the *S. cerevisiae* database (yeastgenome.org).

### Automated strain construction and functionality tests using SGA technology

Strains containing both mCherry and myeGFP were constructed from haploid parent strains of *MATa* and *MATα* mating type, containing myeGFP- or mCherry-tagged genes, by genetic crossing using synthetic genetic array (SGA) technology as described (Tong & Boone, 2007). Functionality tests were performed by directly comparing all strains harboring tagged proteins to their corresponding

deletion strain and to the wild-type strain on 384-well plates. The plates were replica-plated on YPD or high osmolarity media (1 M NaCl) and were then grown either at normal (30°C) or high temperature (37°C).

### Western blotting and antibodies

Yeast cell extracts were prepared using denaturing conditions, as described (Knop *et al*, 1999), and were analyzed by SDS–PAGE using either 4–12% Bis-Tris gels (Life Technologies, NP0323) for proteins < 200 kDa (including tag) or 3–8% Tris-acetate gels (Life Technologies, EA03785) for proteins > 200 kDa. Proteins were transferred to nitrocellulose membranes (Whatman, NBA085C) using semi-dry (< 200 kDa) or tank blotting (> 200 kDa). Detection of myeGFP, mCherry, or the TAP tag was done using specific antibodies (GFP: Miltenyi Biotec, 130-091-833; mCherry: self-made, rabbit-anti-6His-dsred; TAP: Biocat, CAB1001-OB).

A major consideration in FCS experiments is the possibility of a proteolytic cleavage of the fluorescent protein tag. This would lead to a second fluorescent species with different diffusion behavior and would influence the averaged diffusion and co-diffusion measurements. For this reason, we checked by Western blotting all tagged proteins for the presence of lower molecular weight bands that would reflect free GFP or mCherry pools. These experiments showed lower molecular weight bands for some proteins. These were however much less abundant than the respective full-length protein and probably resulted from partial vacuolar proteolysis due to autophagic uptake of the tagged protein. For mCherry-tagged proteins, we always saw lower molecular weight bands, which are likely an *in vitro* artifact resulting from autohydrolysis of the polypeptide via the mCherry chromophore under the conditions used for cell lysis (Gross *et al*, 2000).

### Live cell microscopy

Live cell epifluorescence imaging was performed at room temperature using an Olympus IX81 microscope. Yeast cells were grown in log phase and immobilized in glass-bottomed well chambers (Lab-Tek 155411; Nunc Int., USA; see also next section). GFP fluorescence was recorded using a 470/22 nm excitation filter and a 520/35 nm emission filter. For mCherry-tagged proteins, we used a 566/20 nm excitation filter and a 624/40 nm emission filter.

### FCS/FCCS data acquisition

For general remarks about FCS/FCCS in yeast and a detailed description of the data acquisition and analysis procedures and protocols, see Maeder *et al* (2007). Yeast cells were grown in log phase for at least 16 h and immobilized in concanavalin A (C2010; Sigma, Germany)-pretreated glass-bottomed well chambers (Lab-Tek 155411; Nunc Int.). The chambers were pretreated for at least 15 min with 1% concanavalin A, followed by a wash step with water. All data were recorded on a confocal TCS SP2-FCS system (Leica Microsystems, Wetzlar, Germany) equipped with a 63×1.2 NA water immersion lens. GFP was excited using a 488-nm argon laser at 22 nW, and mCherry was excited by a 561 diode laser at 264 nW. The emitted light was separated by a dichroic mirror (LP560) and then passed into two different detection channels using

the filters BP500–550 (GFP) and HQ638DF75 (mCherry). The duration of each acquisition was 45–60 s, and only one measurement was performed per cell. The pinhole was set to 1.0 airy unit.

### FCS/FCCS data analysis

At the beginning of a measurement session, the observation volume was determined by measuring diffusion times for the fluorescent dyes Alexa 488 (green channel), Alexa 546 (red channel), and Rhodamine Green (cross-correlation channel) at a concentration of 2 nM. The intensity traces collected in the two detection channels were auto- and cross-correlated, and analyzed using custom software. Raw data were autocorrelated by:

$$G_{ij}(\tau) = \frac{<\delta F_i(t)\delta F_j(t+\tau)>}{<\delta F_i(t)><\delta F_j(t)>}$$

with $i = j = 1$, 2 for the autocorrelations in the green and the red channel, respectively, and $i = 1$ for the green channel and $j = 2$ for the red channel in the cross-correlation. $\delta F(t)$ is calculated as the deviation of the present signal intensity $F(t)$ from the mean intensity $<F>$. The brackets indicate an averaging over time. The average of this product for multiple data points is then standardized by the square of the mean intensity, which leads to independence from parameters such as the laser power. The parts of the raw fluctuation traces in which cellular movement or diffusion of vesicles through the observation volume was apparent were either cut out or the file was discarded. A local average approach was used to calculate the autocorrelation function corrected for bleaching of the fluorophores (Im *et al*, 2013). This autocorrelation curve was then fitted to a diffusion model, assuming free diffusion for dyes and anomalous diffusion for *in vivo* data (Wachsmuth *et al*, 2000). The model also corrected for photo-physical effects of the fluorescent proteins and the dyes (triplet-like blinking with a fraction $\Theta$ of the molecules in a non-fluorescent state of lifetime $\tau_{triplet}$):

$$G(\tau) = \frac{1}{N}\left(1 - \Theta + \Theta e^{-\tau/\tau_{triplet}}\right)\left[\left(\frac{\tau}{\tau_{diff}}\right)^{\alpha}\right]^{-1}\left[1 + \frac{1}{k^2}\left(\frac{\tau}{\tau_{diff}}\right)^{\alpha}\right]^{-1/2}$$

At the beginning of each measurement session, the average background fluorescence was estimated using wild-type cells not harboring any fluorescent protein. The background value was then used to correct for the particle number $N$, which was obtained from the fitted auto- or cross-correlation curve. The influence of the background can be described by:

$$G_{bg,corr} = G_{exp}(\tau)\left(\frac{1}{1 - \frac{I_{bg,i}}{I_{total,i}}}\right)\left(\frac{1}{1 - \frac{I_{bg,j}}{I_{total,j}}}\right)$$

The particle numbers $N$ for the red, green, and cross-correlation channels were computationally corrected for bleaching of the fluorophores, background fluorescence, and cross talk between the channels. The particle numbers were converted into concentration values by division through the size of the observation volume. This was determined by calibration measurements with dyes with known diffusion coefficient. Finally, differences in maturation of the

fluorophores and the non-perfect overlap of the two detection volumes were corrected for manually.

Dividing the photon counts per second by the number of particles $N$ yielded the counts per particle per second [cpps].

$$\text{Counts per particle per second [cpps]} = \frac{\text{Photon counts per second}}{\text{Number of particles}}$$

The diffusion time of the molecules $\tau_{diff}$, that is their mean dwell time in the focal volume, is represented as the mean length of the fluctuations. Knowing this time and measuring the lateral diameter $w_{xy}$ of the observation volume allow to calculate the diffusion coefficient $D$ according to:

$$\tau_{diff} = \frac{w_{xy}^2}{4D}$$

The amplitude of the cross-correlation $G_{rg}$ is proportional to the concentration [AB] of complexes found in the observation volume.

$$[AB] = \frac{G_{rg}(0)}{G_{gg}(0)G_{rr}(0)V_{rg}}$$

where $V_{rg}$ is the effective cross-correlation volume.

The relation of the amplitude of cross-correlation and autocorrelation of the two signals can be used as a direct measure of the fraction of one molecule species bound to the complex and can provide insights about the strength of the protein–protein interaction. FCCS reports on protein–protein interactions irrespective on the composition and diversity of complexes that contain the two proteins under investigation. Hence, the dissociation constant $K_D$ must be understood as a measure for the interaction strength under the given condition, which is influenced by the specific concentration of the fluorescently labeled proteins as well as other (sometimes unknown) protein complex members. It is termed therefore apparent or effective $K_D$ ($K_D^{eff}$ (Maeder *et al*, 2007)) and is defined as:

$$K_D^{eff} = \frac{[A][B]}{[AB]}$$

where [A] is the concentration of free green protein, [B] the concentration of free red protein, and [AB] the concentration of proteins found in a complex AB. Protein concentrations measured by FCS include both free protein species as well as the proteins found in complex. Thus, we calculated the $K_D^{eff}$ from our FCS/FCCS data by:

$$K_D^{eff} = \frac{\left([A]_{FCS} - [A]_{FCCS}\right)\left([B]_{FCS} - [B]_{FCCS}\right)}{[AB]_{FCCS}}$$

where $[A]_{FCS}$ and $[B]_{FCS}$ are the concentrations of proteins A and B, respectively, which is detected by FCS in the respective channels, and $[A]_{FCCS}$ and $[B]_{FCCS}$ are the concentrations measured by cross-correlation. A control experiment for a stable interaction (positive control), using the protein Don1 tagged C-terminally to eGFP and N-terminally to 3mCherry, yielded a $K_D^{eff}$ value of 4 nM. A control measurement for no interaction (negative control) was carried out between two proteins which have been shown to

not interact, Don1 and Ste11 (Maeder *et al*, 2007). The $K_D^{\text{eff}}$ value measured for Don1-eGFP, Ste11-3mCherry strain had a value > 1 μM. The dynamic range of cytoplasmic interactions, which could be detected in this study, therefore fell between these two values.

**Supplementary information** for this article is available online: http://msb.embopress.org

## Acknowledgements

We thank Oriol Gallego for contributing plasmids; the ALMF core facility for expert and technical help; Michael Skruzny for help with biochemical assays; and Anton Khmelinskii for comments on the manuscript. Part of this work was funded by a grant from the Deutsche Forschungsgemeinschaft awarded to MKa and MKn (KA3022/1-1).

## Author contributions

MKn and MKa conceived the project and designed together with DB the individual experiments. DB, with help from ST and CG, performed the experiments. DB analyzed the data with help from ST, CG, and MW and interpreted the results with MKn and MKa. MM, ST, and MW contributed reagents, materials, and analysis tools. DB, MKa, and MKn wrote the paper, with input from all authors.

## Conflict of interest

The authors declare that they have no conflict of interest.

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
