## [Review Process File · Molecular Systems Biology]

Quantification of cytosolic interactions identifies Ede1 oligomers as key organizers of endocytosis

Dominik Boeke, Susanne Trautmann, Matthias Meurer, Malte Wachsmuth, Camilla Godlee, Michael Knop and Marko Kaksonen

Corresponding author: Marko Kaksonen, EMBL

Review timeline:

Submission date:	12 May 2014
Editorial Decision:	11 June 2014
Revision received:	19 July 2014
Editorial Decision:	11 August 2014
Revision received:	11 September 2014
Accepted:	01 October 2014

Editor: Maria Polychronidou

Transaction Report:

1st Editorial Decision

11 June 2014

Thank you again for submitting your work to Molecular Systems Biology. We have now heard back from the three referees who agreed to evaluate your manuscript. As you will see from the reports below, the referees acknowledge that the presented findings are potentially interesting. However, they raise a series of concerns, which should be carefully addressed in a revision of the manuscript. The recommendations provided by the reviewers are very clear in this regard.

Reviewer #1:

In the submitted manuscript the authors aim to address several points. They use fluorescence correlation spectroscopy to test many published interactions between the endocytic proteins in the cytoplasm in yeast. This was done by chromosomally tagging proteins with either GFP or mCherry and measuring diffusion rates in the yeast cytosolic compartment. The authors also quantified the relative concentrations of the endocytic proteins, which ranged from low nanomolar (Ark1) to low micromolar (Abp1). Of the 41 interactions tested, 16 were also present in the cytoplasm; however, more interactions could be present that were not in the testable range for this assay.

The authors then turned to self-interacting proteins. Of those tested, only Sla2, Ede1 and Bbc1 exhibited self interactions in the cytoplasm. Ede1 has previously been shown to interact with itself as well as with other components of the endocytic machinery, and is thought to be a scaffold for assembly of the early coat. Ede1 had one of the slowest diffusion coefficients in the FCS experiments, which suggests it is part of a larger complex; therefore, the authors chose to explore

this interaction further. To test this they made several truncations of ede1 fused to a C-terminal GFP tag. Using live cell imaging the authors found that deletion of the coiled coil domain of Ede1 resulted in a loss of cortical localization and instead exhibited a much higher cytoplasmic fluorescence. Interestingly, a mutant that contained only the coiled coil domain could still localize to cortical patches. FCS measurements of constructs without a coiled coil domain showed faster diffusion and higher cytoplasmic concentration, which suggests these proteins can no longer oligomerize. Lastly, the authors wanted to see if they could force homodimerization of ede1 lacking a coiled coil domain using the FRB FKBP rapamycin inducible system. After ten minutes of treatment the artificial ede1 dimers localized to the cortical patches.

Specific Comments

Page 5 Paragraph 2

It is also necessary to look for cleavage products in the western that would indicate the possible presence of free GFP. The diffusion time would be affected if there were degradation products factored into the averages.

Page 7/8

If possible, it would be very useful to the reader if a supplemental table were provided that lists the predicted mass of the monomers vs. the size range measured for a given cytosolic protein complex. This would help collate the information in a way that could facilitate the reader's interpretation of the data and exploration of possible compositions of the complexes.

Figure 2B

- The dotted lines that are connecting the proteins could be prone to misinterpretation. The dashes should be made lighter (e.g., spaced further apart, with each dash rather short), such that they do not accidentally appear as solid lines at first glance.
- Does the number of molecules per cell calculated agree with the number of molecules per cell published on the SGD database? It would be useful to know how much agreement/correlation there is in the two approaches. This may be too complicated (or devoid of meaning?) to calculate, since SGD factors in the total protein levels, while in this study, the focus is on the cytosolic pool, but it might still be an interesting comparison.

Figure 3C

- The cells are difficult to see relative to the fluorescent signal in this part of the figure. Can the contrast or brightness be adjusted on these images?

Figure 4B

- There appeared to be some interesting morphology defects in Figures 4A and 6B. Is this something that was consistently observed, and if so, might this be something that is worth mentioning?

Figure 6

- Missing control - what happens if rapamycin is omitted?
- It would be best to show internalization of a patch with a kymograph, and/or colocalization with a later marker such as Sac6.

Minor Comments

Figure 4B and 5B

- Should label the Y axis as "cytoplasmic concentration"

Page 13 Paragraph 2 Sentence 6

- Whitworth et al. (Traffic, 2014) found that End3 binds to a region of Pan1 that is not phosphorylated. This reference should be included in this section as further support that phosphorylation is unlikely to regulate the association of End3 and Pan1.

Page 3 Paragraph 3 Sentence 2

- The Boettner and D'Agostino citation should be corrected; there are more than two authors on this paper (e.g., Boettner et al.)

Page 3 Paragraph 3 Sentence 3

- Delete "to"

Reviewer #2:

This manuscript reports a systematic study of endocytic factors in yeast. Such factors are recruited sequentially to the plasma membrane, where they orchestrate endocytosis at actin patches. A main question addressed by the authors is which of the interactions among these proteins already occur in the cytosol. To do so, they carry out fluorescence cross-correlation spectroscopy in yeast strains where pairs of each endocytic factor have been tagged by a fluorescent protein. These tagged proteins were generated by homologous recombination and their functionality was thoroughly tested. 16 interactions within the cytosol are observed. A special focus is placed on Ede1 (Eps15), which is shown to oligomerize in the cytosol and to function as a key hub for the nucleation of coats at endocytic sites.

This study pioneer a new direction in the field of endocytosis: the analysis of ensemble of proteins, rather than of one protein at a time. The senior author of this manuscript has been an innovator in the systematic analysis of endocytic factors and this study represents yet another example of his leading the field. Beyond information about protein-protein interactions, this study, which is highly quantitative, reports data on the abundance of all the major endocytic factors. This information will be very useful in the field. The specific new biological insight emerging from this study (new information about Ede1) is limited, but this does not detract from the overall importance of this study in terms of quantitative data and of methodology.

I have only a specific comment. I do not find the puzzling information about the coiled-coil-only Ede1 construct compelling. I suggest to remove it.

Reviewer #3:

This manuscript by Boeke et al. applies fluorescence correlation spectroscopy (FCS) and fluorescence cross-correlation spectroscopy (FCCS) to investigate the cytoplasmic assembly status of 36 components of the yeast endocytic machinery. The results reveal that only a subset of previously characterized interactions occur in the cytosol and also provide evidence that cytoplasmic oligomerization of the early coat component Ede1 is important for an initiation step in endocytosis. The work offers important insights into the process of endocytic coat formation by defining for the first time (to this reviewer's knowledge) the set of interactions that are stable in the cytoplasm *in vivo*. It represents a substantial step towards a full system-wide characterization of the cytoplasmic interaction network of the endocytic machinery and provides a critical foundation for understanding regulation of endocytosis. Overall the data are clear and the interpretations well-founded.

1. As with any imaging technique that relies on fluorescent tags, there is a concern that large tags like triple copies of GFP and mCherry can interfere with protein function. The authors addressed this issue by assaying growth under normal and stress conditions. However, these are general assays that may not be sufficiently sensitive to reveal significant functional defects. For this reason it would be worthwhile to test, at least in a few key cases (Ede1-3myeGFP, Rvs167-3mCherry, Rvs161-3mCherry, Sla1-3mCherry) whether the triple-tagged proteins display the same lifetimes at endocytic sites as single-tagged versions, or the same dynamics in comparison with a standard endocytic component such as Abp1.

2. Figure 6 provides evidence that artificial Ede1 dimerization using FRB/FKBP domains can rescue the Ede1 localization defect caused by deletion of the coiled-coil oligomerization domain. It would be informative for the authors to test whether the FRB/FKBP Ede1 dimers can also rescue the Syp1 localization defect of *ede1* cc mutants. In particular this experiment could help address the authors' model (Fig. 6S, Discussion pg. 16-17) that higher order oligomerization plays an important role in

adaptor clustering.

3. In Figs 1B, C and S1, out of the four subunits of AP-2, Aps2 has a significantly higher concentration and faster diffusion coefficient than the other three subunits. AP-2 is a heterotetramer, and therefore all subunits are expected to display similar properties. The authors should comment on this discrepancy in the text on pg. 6 where slow diffusion of the other AP-2 subunits is addressed. Related to point #1, could the properties observed for Aps2 result from effects of the triple tag that were not apparent in growth assays? Perhaps Aps2 is not efficiently incorporated into the complex.

4. Statistical analysis for significance of differences (or lack thereof) should be provided for bar graphs in Fig. 3B, 4B-D, 6C and S2.

5. In the Results section, page 7, the authors state that "Self-interaction could not be tested for Rvs167 due to problems with the strain" and further refer to the problem at the end of the legend in Figure 2. However, in the Discussion section, page 15, the authors state "... both amphiphysins form a high-affinity heterodimer but not homodimers in the cytoplasm." The authors should modify this conclusion to indicate that their data is consistent with no homodimerization of Rvs161 in the cytosol, but does not allow conclusions about Rvs167 homodimerization.

6. In the methods section, in the data mining segment (page 18), the authors describe how they chose the reported interactions, "taking into account only physical interactions from both high-throughput and manually curated studies". For self-interactions, data from the *Saccharomyces cerevisiae* database (SGD) was used. In the case of Apl3, the self-interaction designation is derived from an SGD interpretation of a single high-throughput study [Babu et al. *Nature* (2012) 489:585-589], from the raw data in which the Apl3-Apl3 interaction is observed in two of the three detergent extractions performed (see wodaklab.org/membrane/links and download Purification data). Babu et al. state in their supplemental methods section (Supplemental page 5, "Calculation of the confidence scores of pairwise protein-protein associations.") that interactions observed in three detergents were given higher confidence scores than the ones found in one or two detergents. The SGD curators based their analysis on the initial cut-off used by Babu et al. However, further statistical analysis was used to define a high confidence interaction network that did not include Apl3 self-interaction (Babu et al. Supplemental table S4 "List of 13343 protein-protein interactions comprising the integrated network, and 1726 interactions comprising the membrane network."). In this context, the absence of Apl3 self-interaction in the cytosol determined by FCCS may reflect the absence of any self-interaction rather than a difference between membrane and cytoplasmic Apl3 self-interaction. For this reason the authors may want to consider removing reference to Apl3 self-interaction or add some discussion of the issues raised here (page 7, 7th line from the bottom; self interacting circle from Figure 2 and Table S5 from Self column).

7. page 3, 5th line from the bottom, "...closely followed by to the appearance...", should read "...closely followed by the appearance..."

8. page 8, 3rd line from the top, "... machinery is in large parts...", should read "... machinery is in large part..."

9. page 9, 8th line from bottom, "μHD" should be "μHD-interacting"

10. pg. 11, 3rd line from bottom of first full paragraph, "...Syp1 was increased in ede1 cc cells as determined by FCS (Fig. 5B)." Fig. 5B displays ede1 but not ede1 cc. This discrepancy should be corrected.

11. Pg. 12, 2nd and 4th lines from the top refer to ede1 cc patches and cite Fig. 6D and 6E but these panels are labeled in the figure and/or legend as ede1 591-1381. This discrepancy should be corrected.

Reviewer #1:

Specific Comments

Page 5 Paragraph 2

It is also necessary to look for cleavage products in the western that would indicate the possible presence of free GFP. The diffusion time would be affected if there were degradation products factored into the averages.

Indeed this is a major worry whenever we do FCS! For this reason we carefully check all tagged proteins for the presence of lower molecular weight bands that would reflect free GFP or mCherry pools. From all our cumulative knowledge we now can say with high confidence that there is no protease in yeast existing in the cytoplasm that would simply cleave off from a fusion protein the particular GFP and mCherry proteins that we use for our work (except for cases where the protein itself is subject to proteolytic processing, of course). Nevertheless, we sometimes observe smaller bands that are however always present in much lower quantities compared to the full-length protein. As it turned out these bands appear to result from partial vacuolar proteolysis, probably due to some autophagic uptake of the tagged protein, especially when the cell culture used for Western blotting was partially saturated (which causes the induction of autophagocytosis). For mCherry we always see smaller bands, but they result from autohydrolysis of the polypeptide via the mCherry chromophore under the acidic and alkaline conditions used for cell lysis and hence represent an in vitro artifact (this is also described in the paper by Gross and Tsien (PNAS, 2000) where the dsRed chromophore is characterized: 10.1073/pnas.97.22.11990; Figure 6).

Page 7/8

If possible, it would be very useful to the reader if a supplemental table were provided that lists the predicted mass of the monomers vs. the size range measured for a given cytosolic protein complex. This would help collate the information in a way that could facilitate the reader's interpretation of the data and exploration of possible compositions of the complexes.

We now provide a plot: Extended view Figure E1 (see also below).

To account for this plot, we introduced the following statement on page 6: "In general, the diffusion coefficients correlated weakly with the molecular weights of the tagged proteins (Fig E1), consistent with the idea that most tagged proteins are not part of large protein assemblies."

Figure 2B

• The dotted lines that are connecting the proteins could be prone to misinterpretation. The dashes should be made lighter (e.g., spaced further apart, with each dash rather short), such that they do not accidentally appear as solid lines at first glance.

We changed the style of the lines.

• Does the number of molecules per cell calculated agree with the number of molecules per cell published on the SGD database? It would be useful to know how much agreement/correlation there is in the two approaches. This may be too complicated (or devoid of meaning?) to calculate, since SGD factors in the total protein levels, while in this study, the focus is on the cytosolic pool, but it might still be an interesting comparison.

The SGD database does list the dataset by Ghaemmaghami et al (2003). In fact there are at present about 9 datasets available from 6 different publications. This data is compiled in the Pax database (pax-DB.org). The log/log plot below shows all values for the proteins in our study. As can be seen, the values in the database easily scatter over 2 orders of magnitude. We think we do not learn anything from this correlation, even more because the pax-DB data sets correspond to total protein/cell values (converted into concentration value), whereas our measurements correspond to the cytoplasmic fraction only. Therefore we do not want to include this plot into the publication, but we provide it here as part of this point-by-point response.

Figure 3C

• The cells are difficult to see relative to the fluorescent signal in this part of the figure. Can the contrast or brightness be adjusted on these images?

All three images were acquired using the same imaging conditions (exposure time etc.) and for visualization the same contrast settings were used. This allows a direct visual comparison of the intensities between the different cells. The low level of soluble Ede1 in the middle panel however

causes that the cell outline is not visible, since most of the protein is present in the bright clusters. To show the cell outlines, we now outline them using dashed red lines. To account for this we correspondingly modified the legend: "Where needed, dashed red lines were used to outline the cell boundaries."

Figure 4B

- *There appeared to be some interesting morphology defects in Figures 4A and 6B. Is this something that was consistently observed, and if so, might this be something that is worth mentioning?*

Indeed, *ede1*-mutants, exhibit morphological defects (e.g. slightly altered cell shapes). This has been described before (Gagny et al. 2000). At one point we considered to use this as readout, but the quantification of this morphological phenotype was not straightforward and initial attempts did not lead to robust results and we therefore did not use it to assess Ede1 functions. We, however, now mention this observation and cite Gagny et al. on page 10.

Figure 6

- *Missing control - what happens if rapamycin is omitted?*

We now included this information in the manuscript: Page 12: "This was not seen when only DMSO was added (Fig. E6)".

- *It would be best to show internalization of a patch with a kymograph, and/or colocalization with a later marker such as Sac6.*

In addition to showing a time series of Sla1 and Ede1 localization we now also provided a kymograph of a patch in Figure 6D.

Minor Comments

Figure 4B and 5B

- *Should label the Y axis as "cytoplasmic concentration"*

We labeled in both Figure panels the Y axes as suggested.

Page 13 Paragraph 2 Sentence 6

- *Whitworth et al. (Traffic, 2014) found that End3 binds to a region of Pan1 that is not phosphorylated. This reference should be included in this section as further support that phosphorylation is unlikely to regulate the association of End3 and Pan1.*

We thank the reviewer for pointing this out, and we added the following sentence to the manuscript (pages 13-14): "A recent study showing that End3 binds to a region in Pan1 that is not phosphorylated provides further support for the conclusion that the interaction between these two proteins is unlikely to be regulated by phosphorylation (Whitworth et al., 2014)."

Page 3 Paragraph 3 Sentence 2

- *The Boettner and D'Agostino citation should be corrected; there are more than two authors on this paper (e.g., Boettner et al.)*

This was corrected.

Page 3 Paragraph 3 Sentence 3

- *Delete "to"*

The error was corrected in the manuscript.

Reviewer #2:

I have only a specific comment. I do not find the puzzling information about the coiled-coil-only Ede1 construct compelling. I suggest to remove it.

The reviewer seems to refer with this suggestion to data shown in Fig 4A and Fig E4, where we show that also the Ede1 coiled-coil domain does localize to foci at the PM. As we show in Figure E4, this localization corresponds to late stages of endocytic vesicles only, and hence suggests that the coiled-coil domain does interact with a component of the late endocytic machinery. Leaving out the coiled-coil-only experiment in Fig 4A, would mislead the reader to think that the coiled-coil does not localize on its own. On the other hand we then need to show (Fig E4) that, although the coiled-coil can localize, it does so only during the late phase. Thus, we need these experiments to explain the behavior of the coiled-coil domain, and the observation of a potential new interaction (with a yet to be identified component) could lead to future insights into the coordination of early and late endocytic events. We therefore do not want to remove this information.

Reviewer #3:

1. As with any imaging technique that relies on fluorescent tags, there is a concern that large tags like triple copies of GFP and mCherry can interfere with protein function. The authors addressed this issue by assaying growth under normal and stress conditions. However, these are general assays that may not be sufficiently sensitive to reveal significant functional defects. For this reason it would be worthwhile to test, at least in a few key cases (Ede1-3myeGFP, Rvs167-3mCherry, Rvs161-3mCherry, Sla1-3mCherry) whether the triple-tagged proteins display the same lifetimes at endocytic sites as single-tagged versions, or the same dynamics in comparison with a standard endocytic component such as Abp1.

According to our experience, most proteins that can withstand a single FP tag can also withstand a triple tag. Nevertheless, we now measured patch lifetimes for Ede1-3myeGFP, Rvs167-3mCherry, Rvs161-3mCherry, Sla1-3myEGFP, Sla2-3myEGFP and compared the values for the corresponding single tagged variants. For the triple tagged Sla1 and Sla2 proteins we could measure statistically significantly increased lifetimes, while the lifetimes of the other proteins did not differ significantly. However, the observed patch lifetimes will depend on our ability to detect the early endocytic sites where Sla1 and Sla2 may be present at low levels. These early stages might be below the detection threshold for the single tags but are visible with triple tags leading to longer apparent lifetimes. Therefore, we think it is difficult to make conclusions about the functionality of the tagged proteins from the lifetime comparison. However, because neither triple tagged Sla1 or Sla2 exhibited a growth phenotype in our assays, while their deletions did, we would like to argue that the conclusions drawn by using these strains are generally valid.

2. Figure 6 provides evidence that artificial Ede1 dimerization using FRB/FKBP domains can rescue the Ede1 localization defect caused by deletion of the coiled-coil oligomerization domain. It would be informative for the authors to test whether the FRB/FKBP Ede1 dimers can also rescue the Syp1 localization defect of ede1 Δ cc mutants. In particular this experiment could help address the authors' model (Fig. 6S, Discussion pg. 16-17) that higher order oligomerization plays an important role in adaptor clustering.

This is indeed a very interesting experiment, and we now tested the effect of artificial dimerization of ede1 Δ cc-FRB-mCherry/ede1 Δ cc-FKBP-mCherry on Syp1-3myEGFP localization. However, we could not detect stable Syp1 patch formation in this strain after addition of rapamycin. This suggests that oligomerization in the wild type (versus dimerization in the artificial construct) of Ede1 molecules may be needed for proper Syp1 localization.

3. In Figs 1B, C and S1, out of the four subunits of AP-2, Aps2 has a significantly higher concentration and faster diffusion coefficient than the other three subunits. AP-2 is a heterotetramer, and therefore all subunits are expected to display similar properties. The authors should comment on this discrepancy in the text on pg. 6 where slow diffusion of the other AP-2

subunits is addressed. Related to point #1, could the properties observed for Aps2 result from effects of the triple tag that were not apparent in growth assays? Perhaps Aps2 is not efficiently incorporated into the complex.

As stated by the reviewer, the cytoplasmic concentration of Aps2 is significantly higher and its diffusion is faster than observed for the other AP-2 complex subunits. The brightness of Aps2-GFP patches is comparable to the brightness of patches of other subunits. We think that the fast diffusion we observe is explained by an excess of free Aps2. We now comment on this point in the manuscript on page 6

4. Statistical analysis for significance of differences (or lack thereof) should be provided for bar graphs in Fig. 3B, 4B-D, 6C and S2.

We added the statistical significances in Fig. 3B, 4B-D and 6C. In Figure S2 (Figure E3 in the revised version), we are aiming to show that the expression levels are similar, although not identical, and thus unlikely to explain the much higher differences seen in the cytosolic concentrations shown in Fig 4B.

5. In the Results section, page 7, the authors state that "Self-interaction could not be tested for Rvs167 due to problems with the strain" and further refer to the problem at the end of the legend in Figure 2. However, in the Discussion section, page 15, the authors state "... both amphiphysins form a high-affinity heterodimer but not homodimers in the cytoplasm." The authors should modify this conclusion to indicate that their data is consistent with no homodimerization of Rvs161 in the cytosol, but does not allow conclusions about Rvs167 homodimerization.

We now include a sentence to explain things further: "Moreover, our data are consistent with no homodimerization of Rvs161 in the cytosol, while homodimerization of Rvs167 could not be tested, due to non-functionality of the Rvs167-3mCherry tag."

6. In the methods section, in the data mining segment (page 18), the authors describe how they chose the reported interactions, "taking into account only physical interactions from both high-throughput and manually curated studies". For self-interactions, data from the Saccharomyces cerevisiae database (SGD) was used. In the case of Apl3, the self-interaction designation is derived from an SGD interpretation of a single high-throughput study [Babu et al. Nature (2012) 489:585-589], from the raw data in which the Apl3-Apl3 interaction is observed in two of the three detergent extractions performed (see wodaklab.org/membrane/links and download Purification data). Babu et al. state in their supplemental methods section (Supplemental page 5, "Calculation of the confidence scores of pairwise protein-protein associations.") that interactions observed in three detergents were given higher confidence scores than the ones found in one or two detergents. The SGD curators based their analysis on the initial cut-off used by Babu et al. However, further statistical analysis was used to define a high confidence interaction network that did not include Apl3 self-interaction (Babu et al. Supplemental table S4 "List of 13343 protein-protein interactions comprising the integrated network, and 1726 interactions comprising the membrane network."). In this context, the absence of Apl3 self-interaction in the cytosol determined by FCCS may reflect the absence of any self-interaction rather than a difference between membrane and cytoplasmic Apl3 self-interaction. For this reason the authors may want to consider removing reference to Apl3 self-interaction or add some discussion of the issues raised here (page 7, 7th line from the bottom; self interacting circle from Figure 2 and Table S5 from Self column).

We added information to the supplementary table indicating that this interaction has a lower confidence score than the other reported interaction.

7. page 3, 5th line from the bottom, "...closely followed by to the appearance...", should read "...closely followed by the appearance..."

Changed

8. page 8, 3rd line from the top, "... machinery is in large parts...", should read "... machinery is in large part..."

Changed

9. page 9, 8th line from bottom, " μ HD" should be " μ HD-interacting"

Changed

10. pg. 11, 3rd line from bottom of first full paragraph, "...Syp1 was increased in *ede1 Δ cc* cells as determined by FCS (Fig. 5B)." Fig. 5B displays *ede1 Δ* but not *ede1 Δ cc*. This discrepancy should be corrected.

It should read "Syp1 was increased in *ede1 Δ* ...". This was changed.

11. Pg. 12, 2nd and 4th lines from the top refer to *ede1 Δ cc* patches and cite Fig. 6D and 6E but these panels are labeled in the figure and/or legend as *ede1 Δ 591-1381*. This discrepancy should be corrected.

It should all read *ede1 Δ 591-1381*. This was changed.

2nd Editorial Decision

11 August 2014

Thank you again for submitting your work to Molecular Systems Biology. We have now heard back from the referee who agreed to evaluate your manuscript. As you will see below, the main concerns raised by the referees have been satisfactorily addressed. However, reviewer #3 refers to a few relatively minor issues, which we would ask you to address in a revision of the manuscript.

Reviewer #3:

Overall the authors have effectively addressed my comments and strengthened the manuscript. This is an excellent manuscript, but I do have remaining questions/comments about the responses to two comments from reviewer #1.

1. The authors respond in their rebuttal to reviewer #1's question about the possible presence of free GFP/Cherry but I did not see anything added to the manuscript to address this important question. If this is the case, I would suggest adding a sentence to the Methods section, pg. 19, at the end of the Western blotting section that lets the reader know that little or no free GFP was detected by western blotting.

2. Reviewer #1 also requested information on the predicted mass of protein monomers and comparison to the measured size. To address this question the authors add a figure showing a plot of predicted mass to diffusion coefficient (Fig. E1). In general the plot supports the authors' conclusion that slow diffusion rates suggest association in a complex. However, Chc1, which is noted as an oligomeric protein with slow diffusion on pg. 5, second paragraph, falls above the fitted line in Fig. E1, meaning that it has a faster diffusion rate relative to the size of the monomer. I understand that this can be explained, but I am concerned about confusing the reader. It might be worthwhile to either address this discrepancy or remove Chc1 as an example. Also, I find the new sentence added at the end of paragraph 2 on pg 6 to potentially conflict with the first sentence of the paragraph. The last sentence suggests that the graph in Fig. E1 is consistent with an absence of "large protein assemblies" while the first sentence states that slow diffusion indicates that the protein is part of a "large complex". The authors could more clearly distinguish between these two terms.

1. *The authors respond in their rebuttal to reviewer #1's question about the possible presence of free GFP/Cherry but I did not see anything added to the manuscript to address this important question. If this is the case, I would suggest adding a sentence to the Methods section, pg. 19, at the end of the Western blotting section that lets the reader know that little or no free GFP was detected by western blotting.*

This is an important point and we have given this quite some consideration throughout the work. We therefore agree with the reviewer and have added the following short paragraph at the proposed location in the Methods section:

A major consideration in FCS experiments is the possibility of a proteolytic cleavage of the fluorescent protein tag. This would lead to a second fluorescent species with different diffusion behavior and would influence the averaged diffusion and co-diffusion measurements. For this reason, we checked by Western blotting all tagged proteins for the presence of lower molecular weight bands that would reflect free GFP or mCherry pools. These experiments showed lower molecular weight bands for some proteins. These were however much less abundant than the respective full-length protein and probably resulting from partial vacuolar proteolysis due to autophagic uptake of the tagged protein. For mCherry-tagged proteins we always saw lower molecular weight bands, which are likely an *in vitro* artifact resulting from autohydrolysis of the polypeptide via the mCherry chromophore under the conditions used for cell lysis (Gross et al., 2000).

2. *Reviewer #1 also requested information on the predicted mass of protein monomers and comparison to the measured size. To address this question the authors add a figure showing a plot of predicted mass to diffusion coefficient (Fig. E1). In general the plot supports the authors' conclusion that slow diffusion rates suggest association in a complex. However, Chc1, which is noted as an oligomeric protein with slow diffusion on pg. 5, second paragraph, falls above the fitted line in Fig. E1, meaning that it has a faster diffusion rate relative to the size of the monomer. I understand that this can be explained, but I am concerned about confusing the reader. It might be worthwhile to either address this discrepancy or remove Chc1 as an example. Also, I find the new sentence added at the end of paragraph 2 on pg 6 to potentially conflict with the first sentence of the paragraph. The last sentence suggests that the graph in Fig. E1 is consistent with an absence of "large protein assemblies" while the first sentence states that slow diffusion indicates that the protein is part of a "large complex". The authors could more clearly distinguish between these two terms.*

The line plotted in Fig E1 is a simple linear fit, illustrating the correlation in our data set. It does not predict the diffusion rates of proteins of different molecular weights nor is it meant to distinguish between monomeric proteins and proteins in larger molecular weight complexes. There are currently no models that could reliably predict the relationship between diffusion rate and molecular mass for proteins in the intracellular environment. In order to not confuse the reader, we now explain the nature of the line in the legend of Figure E1.

In order to emphasize the difference between the new sentence added at the end of paragraph 2 on pg 6 and the first sentence of the paragraph we made it more clear in the first sentence that

We agree that the first sentence could be understood to describe a conclusion: "proteins are in large complexes". However, the first sentence is meant simply to describe a starting assumption, rather than a conclusion. The last sentence summarizes the result "that most tagged proteins are not part of large protein assemblies". To avoid potential confusion between the first and last sentences we modified the first sentence.